# Clinical and nutritional correlates of bacterial diarrhoea aetiology in young children: a secondary cross-sectional analysis of the ABCD trial

Sarah Somji [ORCID],[1,2] Per Ashorn,[1] Karim Manji,[3] Tahmeed Ahmed [ORCID],[4] Md Chisti,[4] Usha Dhingra,[5] Sunil Sazawal,[5] Benson Singa,[5] Judd L Walson [ORCID],[6,7,8] Patricia Pavlinac,[6,7] Naor Bar-Zeev,[9] Eric Houpt,[10] Queen Dube,[11] Karen Kotloff,[12,13] Samba Sow,[14] Mohammad Tahir Yousafzai,[15] Farah Qamar,[15] Rajiv Bahl,[16] Ayesha De Costa,[16] Jonathon Simon,[16] Christopher R Sudfeld,[17] Christopher P Duggan,[17,18] ABCD Study Group

CRS and CPD contributed equally.

For numbered affiliations see end of article.

**Correspondence to**
Ms Sarah Somji; sarahsomji@gmail.com

## ABSTRACT

**Objective** The objective was to assess the association between nutritional and clinical characteristics and quantitative PCR (qPCR)-diagnosis of bacterial diarrhoea in a multicentre cohort of children under 2 years of age with moderate to severe diarrhoea (MSD).

**Design** A secondary cross-sectional analysis of baseline data collected from the AntiBiotics for Children with Diarrhoea trial (NCT03130114).

**Patients** Children with MSD (defined as ≥3 loose stools within 24 hours and presenting with at least one of the following: some/severe dehydration, moderate acute malnutrition (MAM) or severe stunting) enrolled in the ABCD trial and collected stool sample.

**Study period** June 2017–July 2019.

**Interventions** None.

**Main outcome measures** Likely bacterial aetiology of diarrhoea. Secondary outcomes included specific diarrhoea aetiology.

**Results** A total of 6692 children with MSD had qPCR results available and 28% had likely bacterial diarrhoea aetiology. Compared with children with severe stunting, children with MAM (adjusted OR (aOR) (95% CI) 1.56 (1.18 to 2.08)), some/severe dehydration (aOR (95% CI) 1.66 (1.25 to 2.22)) or both (aOR (95% CI) 2.21 (1.61 to 3.06)), had higher odds of having likely bacterial diarrhoea aetiology. Similar trends were noted for stable toxin-enterotoxigenic *Escherichia coli* aetiology. Clinical correlates including fever and prolonged duration of diarrhoea were not associated with likely bacterial aetiology; children with more than six stools in the previous 24 hours had higher odds of likely bacterial diarrhoea (aOR (95% CI) 1.20 (1.05 to 1.36)) compared with those with fewer stools.

**Conclusion** The presence of MAM, dehydration or high stool frequency may be helpful in identifying children with MSD who might benefit from antibiotics.

## INTRODUCTION

Diarrhoea causes over 400 000 deaths annually and strategies to reduce this burden are

### WHAT IS ALREADY KNOWN ON THIS TOPIC

⇒ Nutritional status has a bidirectional relationship with diarrhoea incidence and severity.
⇒ Clinical factors such as dehydration and prolonged duration are known to be a risk factor for severe diarrhoea.
⇒ Limited data exist on the relationship between nutritional and clinical status and specific infectious aetiology of childhood diarrhoea.

### WHAT THIS STUDY ADDS

⇒ Moderate acute malnutrition, dehydration or both were found to have increased odds of likely bacterial aetiology as compared with severe stunting among children with moderate to severe diarrhoea.
⇒ Children with more than six loose or watery stools in the 24 hours before presentation are at higher odds of likely bacterial diarrhoea compared with those with lower stool frequency.

### HOW THIS STUDY MIGHT AFFECT RESEARCH, PRACTICE OR POLICY

⇒ Identifying children at high risk of bacterial diarrhoea may allow a targeted approach with antibiotics during an acute diarrhoeal episode.

informed by studies of diarrhoea aetiology.[1] Bacterial pathogens including enterotoxigenic *Escherichia coli* (ETEC) encoding heat stable toxin (ST-ETEC) and *Shigella* are commonly associated with childhood diarrhoea.[2 3] The WHO recommends antibiotics for diarrhoea only if there is blood in stool or in suspected cases of cholera.[4] Increasing evidence, however, suggests that acute watery diarrhoea can have bacterial (*Shigella* or enteroinvasive *E. coli,* EIEC) aetiology as seen in an age-stratified case–control Global Enteric Multicentre Study (GEMS). This study

enrolled cases who had moderate to severe diarrhoea (MSD) in children under 5 and noted in their collected stool samples that 40.3% *Shigella* spp or EIEC diarrhoea attributed cases were non-dysenteric.[3] Whether children with diarrhoea of bacterial aetiology can be identified, may have implications for antibiotic treatment recommendations.

While the relationship between nutritional status and incidence of diarrhoeal disease is well established, few studies have examined the relationship between child anthropometric and clinical characteristics with the specific aetiology of diarrhoea. Among Bangladeshi children aged 2–5 years with diarrhoea, those with malnutrition (defined as weight-for-age Z-score (WAZ)< −2) were more likely to present with specific parasitic and bacterial enteropathogens specifically ETEC.[5] Another study in Kenya noted that malnourished children were more likely to have MSD due to enteroaggregative *E. coli* (EAEC) but in another study in Bangladesh in a similar cohort, similar children were likely to present with *Shigella*.[6 7] While these studies suggest that acute malnutrition may be an important risk factor for bacterial causes of diarrhoea, they are notable for being performed at single sites, among children with either low-risk diarrhoea or in hospitalised children, or collected with traditional bacterial culture methods, and/or for a limited number of pathogens. We recently completed a large, multicentre clinical trial of antibiotics for MSD in children ages 2–23 months.[8 9] This cohort provided an opportunity to better understand whether nutritional status and other clinical characteristics were associated with specific pathogens causing diarrhoea across several countries. Participants in the trial presented with MSD and were assessed using molecular detection tools for several types of enteropathogens. We hypothesised that among children with MSD, those with moderate acute malnutrition (MAM) would be more likely to have a bacterial aetiology.

## METHODS
### Participants
We studied a subsample of participants from the ABCD trial, a randomised, blinded, placebo-controlled, multi-country trial designed to evaluate whether azithromycin would reduce all-cause mortality in young children with undernutrition and MSD.[8 9] The trial was implemented in Bangladesh, India, Kenya, Malawi, Mali, Pakistan and Tanzania between June 2017 and July 2019 in MSD children 2–23 months of age. For the purpose of this study, MSD was defined as a history of acute watery diarrhoea (caretaker report of ≥3 loose/watery stool in the past 24 hours) for <14 days with one of more of the following: the presence of some/severe dehydration per WHO standards[10] MAM (defined as weight-for-length z-score ≤−2 and >−3 (or mid-upper arm circumference (MUAC) ≥115 mm and <125 mm for children over 6 months)); or severe stunting (length-for-age z-score <−3). Exclusion criteria included receipt of antibiotics currently/in previous

14 days; allergy/contraindication to azithromycin; had dysentery; severe acute malnutrition or clinical suspicion of *Vibrio cholerae* infection; previous/current enrolment of the child in this/any interventional trial; enrolment of a sibling or another child in the same household; or living at a distance from the enrolment centre that would prevent direct observation on days 2 and 3. After written informed consent was obtained, participants were randomised to receive either a 3-day course of azithromycin or placebo.[9] The subsample was participants identified randomly, at baseline with equal numbers from all sites.

### Exposure and confounding variables
MSD defining characteristics were the primary exposure of interest and included severe stunting, MAM, some/severe dehydration or a combination of either of the two or all three characteristics. All anthropometric measurements were performed by two independent trained assessors according to WHO guidelines.[8] Weight was measured using an electronic scale to the closest±10 g, length was obtained using a length board to the nearest 0.1 cm and MUAC was obtained using a non-stretchable tape to the nearest 0.1 cm. Study physicians measured the temperature using a digital thermometer at screening. Fever was calculated as temperature ≥37.5°C. Diarrhoea duration and diarrhoea frequency in the past 24 hours was recorded from mothers recall. Numerous confounding variables were collected at baseline using standardised questionnaires across all sites (online supplemental figure 1). A wealth quintile variable was created using country-specific wealth distribution as reported in each country's most recent Demographic and Health Survey.[11]

### Outcome variables
The primary outcome variable was likely bacterial diarrhoea aetiology assessed by quantitative PCR PCR (qPCR) on a baseline stool sample. Secondary outcomes included specific bacterial diarrhoea aetiology (likely ST-ETEC or *Shigella*). Stool or rectal swab samples were tested by qPCR with a customised 85-target TaqMan array card to determine a cycle threshold value and is detailed elsewhere.[3] Pathogen-specific cut-offs were developed to assign likely diarrhoea aetiology based on the quantity of pathogen DNA/RNA in the stool sample (ie, pathogen burden). These cut-offs[12] were obtained using adapted statistical models from two previously performed large multisite diarrhoea studies: the seven-site GEMS and the eight-site Malnutrition and the Consequences for Child Health and Development cohort study.[3 13] Each cut-off was calculated by taking the median quantity-specific OR from site-specific models from previous studies. Then, the episode-specific attributable fraction (AFe) was calculated. Finally, an LOESS regression was fit and the highest Ct value with ≥0.5 Afe value was picked as the cut-off for each pathogen.[14 15]

For this analysis, Ct values lower than these cut-offs were considered as 'likely' bacterial diarrhoea associated

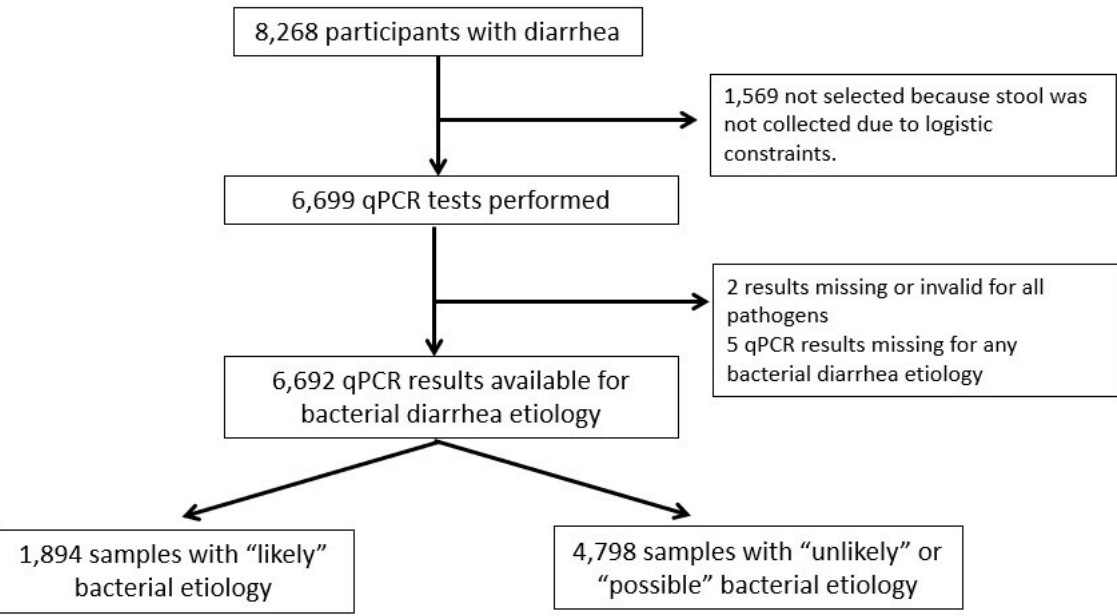

**Figure 1** Flow chart of children aged 2–23 months presenting with moderate to severe diarrhoea. 'Likely' diarrhoea associated aetiology was determined by Ct value cut-offs for specific enteropathogens. Ct values greater than these cut-offs but <35 were considered 'possible' aetiologies while Ct value >35 was considered 'unlikely' aetiology. qPCR, quantitative PCR.

aetiology. Specimens assessed as 'likely' were compared with the other category. Ascribing aetiology to 'likely' with these cut-offs has been previously performed.[16] Bacteria assessed for aetiology and their Ct cut-off values included *Campylobacter* (16.3), typical Enteropathogenic *Escherichia coli* (EPEC) (18.1), ST-ETEC (25.4), *Salmonella* (31.9), *Shigella* (28.7) *and V. cholerae* (32.6). *Shigella* and ST-ETEC were chosen as the main specific aetiologies since they comprised about 84% of all bacterial enteropathogens. Due to logistical and budgetary constraints, stool/rectal swabs from approximately the first 1000 participants at each site were collected. Whole stool sample or flocked rectal swab were stored at −80°C until analysis.

### Statistical analysis

We conducted a cross-sectional analysis of baseline data from the ABCD trial. Logistic regressions were used to examine the association of clinical and anthropometric characteristics, including MSD defining characteristics: fever, diarrhoea duration and stool frequency, with likely bacterial diarrhoea aetiology. For the variable MSD defining characteristic, the presence of severe stunting alone was used as the reference category since participants with severe stunting had the lowest prevalence of likely bacterial diarrhoea. Some previous studies also suggest that stunting is not associated with presence of specific pathogens in stools of children with diarrhoea.[7 17]

Diarrhoea duration at enrolment was categorised as 0–6 days of diarrhoea versus prolonged diarrhoea of 7–14 days, and diarrhoea frequency in the past 24 hours was categorised as low frequency (3–6 stools) versus high frequency (>6 stools). Bivariate (unadjusted) and multivariable analyses were conducted. Multivariable models included covariates for potential confounding factors

that may be associated with both child nutritional status and diarrhoea aetiology based on a literature review. All statistical analyses were performed by using R V.4.0.2 Software (R Foundation for Statistical Computing).

### RESULTS

The parent ABCD trial enrolled 8268 participants who presented with MSD. In total, 6699 had stool/rectal samples collected at baseline and qPCR testing done. Of those, two had missing/invalid qPCR results and five had no bacterial diarrhoea aetiology. On recalculation of anthropometric indexes, seven participants were deemed to have been erroneously enrolled, as they did not meet all enrolment criteria. The deviations from anthropometric enrolment cut-offs were, however, minor and these seven children were retained in the sample. Therefore, 6692 participants across the study sites were included in this analysis (figure 1) and 28% overall had diarrhoea of likely bacterial aetiology (table 1).

We first examined the association of MSD-defining characteristics with likely bacterial aetiology. Compared with children with severe stunting and adjusting for multiple clinical and demographic factors, children presenting with some/severe dehydration only (adjusted OR (aOR) 1.67 (95% CI (1.26 to 2.24)), MAM (aOR 1.57 (95% CI 1.19 to 2.09)), and both MAM and dehydration (aOR 2.23 (95% CI 1.62 to 3.08)) were significantly at higher odds of a likely bacterial infection (table 2). A similar trend was noted for ST-ETEC aetiology where children presenting with dehydration (aOR 1.74 (95% CI 1.19 to 2.62)), MAM (aOR 1.69 (95% CI 1.15 to 2.53)), and MAM and dehydration (aOR 1.83 (95% CI 1.19 to

**Table 1** Baseline characteristics of 6692 mothers and their children aged 2–23 months who presented with moderate-to-severe diarrhoea

| Baseline characteristics | Bangladesh (n=1000) | India (n=998) | Pakistan (n=997) | Kenya (n=1014) | Malawi (n=691) | Mali (n=1000) | Tanzania (n=997) | All sites (n=6692) |
|---|---|---|---|---|---|---|---|---|
| Age (months) | 11.2 (5.0)* | 11.9 (5.6)* | 12.5 (5.5)* | 11.0 (5.7)* | 12.0 (4.8)* | 12.0 (4.6)* | 11.0 (5.2)* | 11.6 (5.3)* |
| Males | 57.8% | 52.9% | 50.6% | 52.7% | 56.9% | 53.9% | 53.2% | 53.8% |
| Any rotavirus vaccination | 1.0% | 2.7% | 7.3% | 99.6% | 99.6% | 97.6% | 99.9% | 60% |
| Exclusively breastfed at time of enrolment | 1.7% | 6.9% | 9.2% | 17.9% | 4.1% | 5.4% | 14.4% | 8.7% |
| Maternal age, years | 24 (5) | 26 (4) | 27 (6) | 26 (6) | 25 (6) | 26 (6) | 27 (6) | 26 (6) |
| Maternal body mass index, kg/m² | 22.3 (4.3) | 22.6 (4.3) | 23.2 (5.0) | 23.2 (3.8) | 23.1 (5.3) | 24.0 (4.9) | 24.5 (5.0) | 23.3 (4.7) |
| Maternal education, completed school years | 5.8 (4.0) | 4.8 (4.8) | 2.9 (4.0) | 9.0 (2.9) | 8.3 (2.8) | 4.3 (5.0) | 7.8 (2.7) | 6.0 (4.4) |
| Number of children <5 years of age in house | 1.2 (0.4) | 1.9 (0.9) | 2.0 (1.0) | 1.7 (0.8) | 1.4 (0.6) | 2.4 (1.8) | 1.3 (0.5) | 1.7 (1.1#) |
| Fever (defined as temperature >37.5°C) | 15.3% | 7.6% | 9.0% | 6.0% | 19.4% | 16.9% | 12.5% | 12.1% |
| Some/severe dehydration | 25.1% | 48.4% | 28.5% | 91.8% | 77.7% | 23.1% | 92.8% | 54.4% |
| Length for age z-score (LAZ) | −1.83 (1.15) | −2.12 (1.28) | −2.15 (1.28) | −0.81 (1.26) | −1.55 (1.47) | −1.16 (1.13) | −0.76 (1.14) | −1.48 (1.36) |
| Weight for length z-score (WLZ) | −1.69 (0.90) | −1.37 (1.01) | −1.45 (0.96) | −0.28 (1.19) | −0.62 (1.19) | −2.04 (0.74) | −0.39 (1.27) | −1.14 (1.23) |
| Severe stunting† | 19.5% | 27.6% | 27.5% | 5.6% | 16.5% | 6.5% | 2.2% | 15.0% |
| Moderate acute malnutrition‡ | 73.6% | 57.3% | 70.6% | 14.0% | 26.0% | 85.0% | 16.1% | 49.9% |
| Presence§ of any bacterial pathogen¶ in stool | 36.3% | 27.5% | 30.9% | 21.5% | 31.7% | 28.6% | 22.9% | 28.3% |
| Presence¶ of Shigella in stool | 12.8% | 16.0% | 18.2% | 7.0% | 11.3% | 13.5% | 9.2% | 12.6% |
| Presence¶ of ST-ETEC** in stool | 19.4% | 9.9% | 12.7% | 10.4% | 20.4% | 11.9% | 10.7% | 13.3% |

*Numbers indicate percentages for categorical variable and mean (SD) for continuous variables.
†Defined as LAZ of <−3.0.
‡Defined as WLZ<−2.0 and ≥−3.00 or mid-upper arm circumference of <12.5 and ≥11.5 if participant was 6 months or older.
§Indicated as a Cq value below relevant cut-off in a quantitative PCR (Liu et al).[3]
¶Includes the following bacteria—Cq cut-off for diagnostic quantity to be termed as associated with diarrhoea: Campylobacter spp (16.3), typicalEnteropathogenic Escherichia coli EPEC (18.1), ST-ETEC (25.4), Salmonella spp (31.9), Shigella spp (28.7), Vibrio cholerae (32.6).
**(Heat stable Toxin) STh-producing enterotoxigenic Escherichia coli.
ST-ETEC, stable toxin-enterotoxigenic Escherichia coli.

**Table 2** Clinical and nutritional correlates of likely bacterial aetiology of diarrhoea among children aged 2–23 months with moderate-to-severe diarrhoea adjusted for factors

| | Bacterial aetiology | | |
| --- | --- | --- | --- |
| | Proportion n/N (%) | Adjusted* OR (95% CI) | P value |
| Moderate to severe diarrhoea defining characteristics | | | |
| Severe stunting only | 94/417 (22.5) | Ref. | – |
| Some/severe dehydration only | 764/2832 (27.0) | 1.67 (1.26 to 2.24) | <0.001 |
| MAM only | 675/2228 (30.3) | 1.57 (1.19 to 2.09) | 0.002 |
| MAM and some/severe dehydration | 209/626 (33.4) | 2.23 (1.62 to 3.08) | <0.001 |
| MAM and severe stunting | 109/408 (26.7) | 1.13 (0.79 to 1.62) | 0.52 |
| Some/severe dehydration and severe stunting | 22/95 (23.2) | 1.42 (0.79 to 2.49) | 0.23 |
| MAM, some/severe dehydration and severe stunting | 20/84 (23.8) | 0.80 (0.39 to 1.53) | 0.52 |
| Fever | | | |
| No | 1646/5884 (28.0) | Ref. | – |
| Yes | 248/808 (30.7) | 1.11 (0.93 to 1.32) | 0.26 |
| Duration of diarrhoea (excluding day of enrolment) | | | |
| Diarrhoea (0–6 days) | 1781/6328 (28.1) | Ref. | – |
| Prolonged duration (7–14 days) | 113/364 (31.0) | 1.22 (0.94 to 1.58) | 0.13 |
| Frequency of diarrhoea in the past 24 hours | | | |
| Low frequency (3–6 stools) | 912/3536 (25.8) | Ref. | – |
| High frequency (>6 loose stools) | 982/3156 (31.1) | 1.20 (1.05 to 1.36) | 0.007 |

*Adjusted for child age, child sex, household wealth quintile, maternal age, maternal education, country of enrolment, rotavirus vaccination status, maternal BMI, total number of children under age 5 year in the household, defining characteristic of moderate to severe diarrhoea, fever, duration of diarrhoea and high versus low frequency of stools in past 24 hours.
BMI, body mass index; MAM, moderate acute malnutrition.

2.89)) compared with those severely stunted, were at higher odds of a likely ST-ETEC aetiology (table 3). For *Shigella*, only children presenting with MAM and dehydration were found to have higher odds (aOR 1.64 (95% CI 1.09 to 2.50)) compared with those severely stunted (table 3). In bivariate associations of likely bacterial aetiology, MAM and dehydration increased odd of likely bacterial aetiology (OR 1.70 (95% CI 1.28 to 2.27)) and MAM increased the odds of likely ST-ETEC aetiology (OR 1.58 (95% CI 1.13 to 2.26)) versus severe stunting (online supplemental table 1).

Fever and duration of diarrhoea were not associated with bacterial aetiology or *Shigella* or ST-ETEC aetiology (tables 2 and 3). However, children with high stool frequency (>6 stools) had increased odds of likely bacterial aetiology (aOR 1.20 (95% CI 1.05 to 1.36)) compared with those with lower stool frequency.

Since the qPCR analysis often detected more than one type of enteropathogen, we performed sensitivity analyses to evaluate the association between clinical and anthropometric characteristics with likely bacterial diarrhoea aetiology in children where only the enteropathogen of interest was found. The proportion of children who had only a single likely diarrhoea bacterial aetiology was 1110/6648 (16.7%). No statistically significant associations were observed between anthropometrics and specific likely bacterial pathogens in these sensitivity

analyses. High stool frequency, however, was associated with higher odds of likely bacterial diarrhoea aetiology (aOR 1.23 (95% CI 1.05 to 1.45)), and *Shigella* aetiology (aOR 1.32 (95% CI 1.04 to 1.68)) after adjusting for multiple factors (online supplemental tables 2 and 3).

## DISCUSSION

In this analysis of 6692 children with MSD whose aetiology was assessed by qPCR methods, we found that several anthropometric and clinical factors were associated with likely bacterial diarrhoea aetiology. Children presenting with either MAM, dehydration or a combination of both, had higher odds of a likely bacterial and ST-ETEC diarrhoea aetiology compared with severely stunted children. Children with MAM and dehydration were also more likely to have *Shigella* diarrhoea aetiology compared with severely stunted children. Children with high stool frequency had higher odds of likely bacterial diarrhoea aetiology, a finding confirmed on sensitivity analysis evaluating children with only a single pathogen identified.

Our findings of association between MAM and a likely bacterial and ST-ETEC diarrhoea aetiology are consistent with some earlier studies.[7 18] Tickell *et al* studied 1363 Kenyan children aged 6–59 months who presented with acute diarrhoea. The study noted that children with

**Table 3** Clinical and nutritional correlates of likely *Escherichia coli* encoding heat-stable ST-ETEC and *Shigella* aetiology of diarrhoea among children aged 2–23 months with moderate-to-severe diarrhoea adjusted for factors

| | *Escherichia coli* encoding heat-stable toxin aetiology | | | *Shigella* aetiology | | |
|---|---|---|---|---|---|---|
| | Proportion n/N (%) | Adjusted* OR (95% CI) | P value | Proportion n/N (%) | Adjusted* OR (95% CI) | P value |
| Moderate to severe diarrhoea defining characteristics | | | | | | |
| Severe stunting only | 40/417 (9.6) | Ref. | – | 52/417 (12.5) | Ref. | – |
| Some/severe dehydration only | 378/2815 (13.4) | 1.74 (1.19, 2.62) | 0.006 | 305/2832 (10.8) | 1.18 (0.82 to 1.73) | 0.38 |
| MAM only | 319/2227 (14.3) | 1.69 (1.15 o to 2.53) | 0.009 | 320/2224 (14.4) | 1.29 (0.90 to 1.86) | 0.17 |
| MAM and some/severe dehydration | 81/623 (13.0) | 1.83 (1.18 to 2.89) | 0.008 | 90/626 (14.4) | 1.64 (1.09 to 2.50) | 0.02 |
| MAM and severe stunting | 45/408 (11.0) | 1.04 (0.62 to 1.74) | 0.87 | 63/408 (15.4) | 1.08 (0.69 to 1.70) | 0.75 |
| Some/severe dehydration and severe stunting | 14/95 (14.7) | 1.92 (0.92 to 3.82) | 0.07 | 5/95 (5.3) | 0.52 (0.17 to 1.27) | 0.19 |
| MAM, some/severe dehydration and severe stunting | 12/84 (14.3) | 1.19 (0.46 to 2.71) | 0.69 | 10/84 (11.9) | 0.68 (0.25, 1.58) | 0.41 |
| Fever | | | | | | |
| No | 767/5865 (13.1) | Ref | – | 736/5883 (12.5) | Ref | – |
| Yes | 122/806 (15.1) | 1.12 (0.89 to 1.40) | 0.31 | 109/808 (13.5) | 1.15 (0.90 to 1.46) | 0.26 |
| Duration of diarrhoea (excluding day of enrolment) | | | | | | |
| Diarrhoea (0–6 days) | 846/6308 (13.4) | Ref | – | 783/6327 (12.4) | Ref | – |
| Prolonged duration (7–14 days) | 43/363 (11.8) | 1.05 (0.72 to 1.50) | 0.79 | 62/364 (17.0) | 1.29 (0.92 to 1.78) | 0.13 |
| Frequency of diarrhoea in the past 24 hours | | | | | | |
| Low frequency (3–6 stools) | 433/3521 (12.3) | Ref | – | 406/3535 (11.5) | Ref | – |
| High frequency (>6 loose stools) | 456/3150 (14.5) | 1.10 (0.93 to 1.31) | 0.28 | 439/3156 (13.9) | 1.17 (0.98 to 1.40) | 0.08 |

*Adjusted for child age, child sex, household wealth quintile, maternal age, maternal education, country of enrolment, rotavirus vaccination status, maternal BMI, total number of children under age 5 year in the household, defining characteristic of moderate to severe diarrhoea, fever, duration of diarrhoea and high versus low frequency of stools in past 24 hours.
BMI, body mass index; MAM, moderate acute malnutrition; ST-ETEC, stable toxin-enterotoxigenic *Escherichia coli*.

acute diarrhoea and wasting (defined as MUAC<12.5 cm) were 1.8 times more likely to have EAEC diarrhoea aetiology compared with those not wasted. Although the definition of acute malnutrition was different from our definition, they used similar PCR-based automated bacterial pathogen detection methods. A case–control study among children aged 6–23 months in Bangladesh[18] noted that bacterial enteropathogens such as *Shigella* and EAEC were more common among children who were underweight (defined as WAZ<−2), a finding similar to ours. This study, however, enrolled children with malnutrition and only ~5% of those enrolled had diarrhoea at baseline, whereas in our study all children had MSD. Similarly, a recent cross-sectional study carried out in Ethiopia noted that being wasted was associated with Salmonella infection.[19]

Children with some/severe dehydration, with/without MAM, had greater odds of likely bacterial or ST-ETEC diarrhoea aetiology compared with children with severe stunting. Some/severe dehydration is known to be a risk factor for severe diarrhoea (and even death) in children but not usually associated with bacterial diarrhoea.[20 21] In a study carried out in chidren <5 years presenting to a

health clinic in Rwanda, *Shigella* was not associated with severe dehydration.[22] Moreover, in a retrospective study carried out in Lao PDR to evaluate clinical and aetiology associated with severity of acute diarrhoea, among hospitalised children, 48.4% presented with dehydration and prevalence of dehydration was higher in those with rotavirus aetiology versus rotavirus negative children. This association of dehydration was not noted in children with *Shigella* or *Salmonella* aetiology, although the low numbers of samples positive for *Salmonella* (n=4) and *Shigella* (n=1) vs rotavirus (n=10) is noteworthy.[23] Neither of the studies noted the association of dehydration in any bacterial diarrhoea in contrast with our findings.

A reanalysis of the ABCD trial including molecular diagnostics was conducted to understand the effect of azithromycin in the outcomes in children with and without bacterial diarrhoea aetiology.[16] It was noted that among those with likely and possible bacterial aetiology, those in the azithromycin arm had lower risk of diarrhoea at 3 days after randomisation in comparison to those in the placebo arm. Additionally, within the 90-day follow-up period, azithromycin use was associated with lower risk of deaths/hospitalisations in children with likely and

possible bacterial diarrhoea. As noted in Pavlinac *et al*'s results [12] and previous studies, bacterial diarrhoea may not always present with clinical signs and symptoms of dysentery[13 24] and, therefore, limiting antibiotics to those with clinically apparent dysentery may limit treatment of bacterial diarrhoea. The findings from our analysis outlines clinical and nutritional risk factors to potentially identify such high+risk children who may benefit from antibiotics before culture results are known.

Our secondary analysis has some limitations. First, the observational, cross-sectional nature of the analysis limits the ability to make causal inferences. Second, we may not have had adequate power to detect more moderate magnitudes of association between nutritional status and diarrhoea aetiology. Third, our study included children with MSD which may limit the generalisability of our findings. We note that lack of a non-malnourished group (or severely malnourished group such as SAM) adds to the limited of generalisability. Finally, the use of enteropathogens cut-offs is complicated in settings where asymptomatic carriage of enteropathogens is high. It is noteworthy that these cut-offs were derived from previous case–control studies, thus may be subject to further refinement. It is also noted that there were a small number of children with individual pathogens which limited the power to detect differences between groups. Strengths of the study include the multicountry setting, the prospective nature of the data collection, the inclusion of numerous possible confounding variables, standardised measures of anthropometric data and the use of novel enteropathogen detection tools. We also note that additional covariates that could influence the associations such as preterm birth, bloody diarrhoea, severe wasting were not included since some were not measured and others were exclusion criteria.

In summary, in this large cohort of children with MSD, we note that MAM and dehydration as well as high frequency of stools were associated with an increased odd of likely bacterial aetiology, compared with those with severe stunting. These findings suggest that these characteristics could be helpful criteria for children presenting with MSD who might benefit from antibiotics during an acute diarrhoeal episode. Future studies could examine other, more specific risk factors for diarrhoea aetiology, including micronutrient status,[25 26] exposure to food and water sources,[27–29] vaccine status[30] and others. Additional trials of the utility of targeted antibiotic therapy also seem warranted.

**Author affiliations**
[1]Center for Child, Adolescent, and Maternal Health Research, Faculty of Medicine and Health Technology, Tampere University, Tampere, Finland
[2]Department of Epidemiology, Muhimbili University of Health and Allied Sciences, Dar es Salaam, Tanzania, United Republic of
[3]Department of Pediatrics, Muhimbili University of Health and Allied Sciences, Dar es Salaam, Tanzania, United Republic of
[4]Division of Nutrition and Clinical Sciences, International Centre for Diarrhoeal Disease Research, Dhaka, Bangladesh
[5]Centre for Public Health Kinetics (CPHK), Delhi, India
[6]Department of Global Health, University of Washington, Seattle, Washington, USA
[7]Department of Epidemiology, University of Washington, Seattle, Washington, USA
[8]Department of Pediatrics and Medicine (Infectious Diseases), University of Washington, Seattle, Washington, USA
[9]International Vaccine Access Center, Johns Hopkins Bloomberg School of Public Health, Baltimore, Maryland, USA
[10]Department of Medicine, Infectious Diseases, University of Virginia, Charlottesville, Virginia, USA
[11]Department of Pediatrics, Queen Elizabeth Central Hospital, Blantyre, Southern Region, Malawi
[12]Center for Vaccine Development and Global Health, University of Maryland Baltimore, Baltimore, Maryland, USA
[13]Department of Pediatrics, Center for Vaccine Development and Global Health, University of Maryland, College Park, Maryland, USA
[14]Centre pour le Developpement des Vaccins Mali, Bamako, Mali
[15]Department of Pediatrics and Child Heath, The Aga Khan University, Karachi, Sindh, Pakistan
[16]Department of Maternal, Child, and Adolescent Health and Aging, World Health Organization, Geneva, Switzerland
[17]Department of Global Health and Population and Nutrition, Harvard University T H Chan School of Public Health, Boston, Massachusetts, USA
[18]Division of Gastroenterology, Hepatology and Nutrition, Boston Children's Hospital, Boston, Massachusetts, USA

**Acknowledgements** We would like to thank the families and children who participated in this trial as well as the staff who helped carried out the trial in each of the seven sites. We also thank additional members of the ABCD Study Group, Bangladesh.

**Collaborators** ABCD Study Group: Muhammad Waliur Rahman, Irin Parvin, Md. Farhad Kabir, Pratibha Dhingra, Arup Dutta, Anil Kumar Sharma, Vijay Kumar Jaiswal, Churchil Nyabinda, Christine McGrath, Emily L Deichsel, Maurine Anyango, Kevin Mwangi Kariuki, Doreen Rwigi, Stephanie N Tornberg-Belanger, Fadima Cheick Haidara, Flanon Coulibaly, Jasnehta Permala-Booth, Dramane Malle, Nigel Cunliffe, Latif Ndeketa, Desiree Witte, Chifundo Ndamala, Shahida Qureshi, Sadia Shakoor, Rozina Thobani, Jan Mohammed, Rodrick Kisenge, Mohamed Bakari, Cecylia Msemwa, Abraham Samma, James Platts-Mills, Jie Liu.

**Contributors** SSomji, PA, KM, CPD and CRS: conceptualisation. All authors: approval and validation of concept. SSomji, PA, PP, CPD and CRS: methodology. SSomji and CRS: analysis. SSomji and CPD: writing–original draft. all authors: Critical review, comments and edits of paper. PA, KM and CPD: supervision. ABCD Study group: data collection, supervision and processing of parent trial. SSomji: guarantor.

**Funding** The parent trial was funded by the Bill & Melinda Gates Foundation (grant OPP 1126331). Under the grant conditions of the Foundation, a Creative Commons Attribution 4.0 Generic Licence has already been assigned to the Author Accepted Manuscript version that might arise from this submission. CD was supported, in part, by the National Institutes of Health (K24 DK104676 and P30 DK040561).

**Disclaimer** The study sponsor had no role in the analysis, interpretation of results, writing of this article or decision to submit this paper for publication.

**Competing interests** None declared.

**Patient and public involvement** Patients and/or the public were not involved in the design, or conduct, or reporting, or dissemination plans of this research.

**Patient consent for publication** Not applicable.

**Ethics approval** This study involves human participants and the ABCD trial ( ClinicalTrials.gov Identifier: NCT03130114) was approved by the WHO Ethics Review Committee (ERC.0002722) and the institutional review boards of all participating sites; Bangladesh (ERC.0002904) (International Centre for Diarrhoeal Disease Research), India (ERC.0002912) (Institutional Ethics Committee, Subharti Medical College and Hospital, Swami Vivekan and Subharti University), Kenya (ERC.0002932) (Kenya Medical Research Institute Ethical Review Committee and the University of Washington Institutional Review Board), Malawi (ERC.0002922) (University of Malawi College of Medicine Research Ethics Committee), Mali (ERC.0002916) (Comite d' Ethiquedela FMPOS (ERCatUSTTB)), Pakistan (ERC.0002948) (Aga Khan University, Ethical Review Committee), Tanzania (ERC.0002914) (Muhimbili University of Health and Allied Sciences Senate Research and Publications Committee; Tanzanian Food and Drug Administration;

National Institute for Medical Research), and United States (Boston Children's Hospital Institutional Review Board; University of Maryland, Baltimore Research Ethics Committee). Participants gave informed consent to participate in the study before taking part.

**Provenance and peer review**  Not commissioned; externally peer reviewed.

**Data availability statement**  Data are available on reasonable request.

**ORCID iDs**
Sarah Somji http://orcid.org/0000-0001-6031-830X
Tahmeed Ahmed http://orcid.org/0000-0002-4607-7439
Judd L Walson http://orcid.org/0000-0003-4836-720X

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
