## [Reviewer comments · BMJ Paediatrics Open]

ARTICLE DETAILS

TITLE (PROVISIONAL)	Clinical and nutritional correlates of bacterial diarrhea etiology in young children: A secondary cross-sectional analysis of the ABCD trial
AUTHORS	Somji, Sarah Ashorn, Per Manji, Karim Ahmed, Tahmeed Chisti, Md. Dhingra, Usha Sazawal, Sunil Singa, Benson Walson, Judd L. Pavlinac, Patricia Bar-Zeev, Naor Haupt, Eric Dube, Queen Kotloff, Karen Sow, Samba Yousafzai, Mohammad Tahir Qamar, Farah Bahl, Rajiv De Costa, Ayesha Simon, Jonathon Sudfeld, Christopher Duggan, Christopher Study Group, ABCD

VERSION 1 – REVIEW

REVIEWER	Dr. Gilbert Sterling Octavius Pelita Harapan University Faculty of Medicine
REVIEW RETURNED	24-Jan-2024

GENERAL COMMENTS	Dear Authors, I have read a secondary analysis from the ABCD trial on clinical and nutritional correlates of bacterial diarrhea etiology in young children. Although this is an interesting review, several issues need to be addressed: 1) Under the heading "How this study might affect research," it seems like the sentence is abruptly cut-off 2) there is an awkward gap between ST-ETEC and Shigella in the introduction. 3) How was the Ct value checked? How was the PCR conducted?
---

	4) What was the cut-off Ct value? It was listed in Figure 1 but not in the methodology section 5) "For this analysis, Ct values greater than these cut-offs were considered as not diarrhea-associated etiology." I believe it should be a bacteria-associated etiology. 6) ".....nutritional status and diarrhea etiology based on a literature review." What are the factors that have been identified in the literature review? 7) Who assessed the dehydration? Dehydration in children is notoriously difficult to ascertain completely with just physical examinations, which may introduce bias.
--	---

VERSION 1 – AUTHOR RESPONSE

Reviewer: 1

Dr. Gilbert Sterling Octavius, Pelita Harapan University Faculty of Medicine

Comments to the Author

Dear Authors,

I have read a secondary analysis from the ABCD trial on clinical and nutritional correlates of bacterial diarrhea etiology in young children. Although this is an interesting review, several issues need to be addressed:

1) Under the heading "How this study might affect research," it seems like the sentence is abruptly cut-off

Thank you for noting this. The sentence is now complete and reads as follows:

“Identifying children at high risk of bacterial diarrhea may allow a targeted approach with antibiotics during an acute diarrheal episode.”

2) there is an awkward gap between ST-EPEC and Shigella in the introduction.

Thank you for this comment. We have added an “and” between the two words to avoid the awkward gap.

3) How was the Ct value checked? How was the PCR conducted?

We defer to the editor if more detail is required than we have described as follows:

“Stool or rectal swab samples were tested by qPCR with a customized 85-target TaqMan array card to determine a cycle threshold (CTt) value and is detailed elsewhere (Liu et al., 2016). Pathogen-specific cutoffs were developed to assign likely diarrhea etiology based on the quantity of pathogen DNA/RNA in the stool sample (i.e., pathogen burden). These cut-offs(Pavlinac et al., n.d.) were obtained using

adapted statistical models from two previously performed large multisite diarrhea studies: the seven-site Global Enteric Multicenter Study (GEMS) and the eight-site Malnutrition and the Consequences for Child Health and Development (MAL-ED) cohort study (Liu et al., 2016; Platts-Mills et al., 2018). Each cut-off was calculated by taking the median quantity-specific odds ratio from site specific models from previous studies. Then, the episode-specific attributable fraction (AF_e) was calculated. Finally, a LOESS regression was fit and the highest Ct value with ≥ 0.5 Afe value was picked as the cut-off for each pathogen (Kotloff et al., 2012; The MAL-ED Network Investigators, 2014). “

4) What was the cut-off Ct value? It was listed in Figure 1 but not in the methodology section

Thank you for this comment. The Ct Values have been added in the following section:

“Bacteria assessed for etiology and their Ct cut-off values included Campylobacter (16.3), tEPEC (18.1), ST-EPEC (25.4), Salmonella (31.9), Shigella (28.7), and V.cholerae (32.6).”

5) "For this analysis, Ct values greater than these cut-offs were considered as not diarrhea-associated etiology." I believe it should be a bacteria-associated etiology.

We appreciate your feedback. The sentence has been edited and now reads as follows:

“For this analysis, Ct values lower than these cut-offs were considered as bacterial diarrhea associated etiology.”

6) ".....nutritional status and diarrhea etiology based on a literature review." What are the factors that have been identified in the literature review?

Thank you for noting this. Those factors are detailed in the exposure and confounding variable section. They are as follows: “child age, child sex, household wealth quintile, maternal age, maternal education, country of enrolment, rotavirus vaccination status, maternal BMI, total number of children under age 5 year in the household, defining characteristic of moderate to severe diarrhea, fever, duration of diarrhea, and high vs. low frequency of stools in past 24h.”

7) Who assessed the dehydration? Dehydration in children is notoriously difficult to ascertain completely with just physical examinations, which may introduce bias.

Our experienced study clinicians evaluated dehydration according to WHO standards. We have amended this text as follows: “...the presence of some/severe dehydration per WHO standards.”